# Increased slow dynamics defines ligandability of BTB domains

Vladlena Kharchenko[1,4], Brian M. Linhares[2,3,4], Megan Borregard[2], Iwona Czaban [1], Jolanta Grembecka [2], Mariusz Jaremko[1], Tomasz Cierpicki [2] ✉ & Łukasz Jaremko [1] ✉

Efficient determination of protein ligandability, or the propensity to bind small-molecules, would greatly facilitate drug development for novel targets. Ligandability is currently assessed using computational methods that typically consider the static structural properties of putative binding sites or by experimental fragment screening. Here, we evaluate ligandability of conserved BTB domains from the cancer-relevant proteins LRF, KAISO, and MIZ1. Using fragment screening, we discover that MIZ1 binds multiple ligands. However, no ligands are uncovered for the structurally related KAISO or LRF. To understand the principles governing ligand-binding by BTB domains, we perform comprehensive NMR-based dynamics studies and find that only the MIZ1 BTB domain exhibits backbone μs-ms time scale motions. Interestingly, residues with elevated dynamics correspond to the binding site of fragment hits and recently defined HUWE1 interaction site. Our data argue that examining protein dynamics using NMR can contribute to identification of cryptic binding sites, and may support prediction of the ligandability of novel challenging targets.

Predictions of protein ligandability and/or druggability can tremendously facilitate the development of both chemical probes and novel drug candidates. Ligandability refers to the proclivity of a protein target to bind small molecules with high affinity, whereas druggability reflects the feasibility of developing potent and safe molecules with therapeutic efficacy[1]. The ligandability, and further, druggability, of a given protein depends on the physicochemical and topological properties of small-molecule binding sites[2]. Known tractable drug targets, for example, GPCRs, ion channels, and kinases[3,4], generally present relatively small binding sites, with Solvent Accessible Surface Areas (SASA) < 1000 $A^2$, and well-defined binding pockets[2,5]. Conversely, less ligandable targets often present binding sites that are larger (SASAs > 1000 $A^2$)[6-8], with smaller radii of curvature[2]. These characteristics reflect larger, "flatter", and less topologically-defined interfaces for protein–protein interactions (PPIs) and are consequently more challenging targets for drug discovery[9-12].

Various computational approaches have been employed to explore the ligandability of protein targets, typically based on static structural data and topological properties of putative binding sites[13-18]. These methods provide reasonable predictions of ligandability for enzymes and receptors. However, challenging targets necessitate new approaches, particularly when it comes to PPIs. For example, the presence of ligand binding hot spots can be predicted based on static crystal structures employing FTMAP software[19]. The presence of protein dynamics can be accounted by simultaneous analysis of multiple crystal structures[20], inspecting the electron density map alterations[21], or by analyzing the conformational heterogeneity from the molecular dynamics simulation data sets[22]. Additionally, ligandability can be evaluated using experimental methods, and fragment-based screening (FBS) has proven to be a particularly valuable approach for this task. Assessment of protein ligandability by FBS was first developed by Hajduk et al., and is based on both the rate of hits from FBS, and on the

[1]Smart Health Initiative (SHI), Red Sea Research Center (RSRC), Bioscience Program, Biological and Environmental Science & Engineering (BESE), King Abdullah University of Science and Technology (KAUST), 23955-6900 Thuwal, Saudi Arabia. [2]Department of Pathology, University of Michigan, 1150 West Medical Center Dr, MSRB I, Room 4510D, Ann Arbor, MI 48108, USA. [3]Present address: Siduma Therapeutics, Inc., 55 Church St., 8th Fl., New Haven, CT 06510, USA. [4]These authors contributed equally: Vladlena Kharchenko, Brian M. Linhares. ✉e-mail: tomaszc@umich.edu; lukasz.jaremko@kaust.edu.sa

physicochemical properties of binding sites[14,23]. FBS was demonstrated to be highly efficient in predicting ligandability and developing optimized inhibitors, particularly in the case of PPI interfaces[24]. AstraZeneca proposed a more rigorous approach to predict protein ligandability based on FBS, by combining screen hit rates and affinities of identified ligands[25].

BTB domains are common PPI motifs present in transcription factors and epigenetic scaffolding proteins[26,27]. BTB domain-containing proteins have emerged as pharmacological targets and include BCL6, an oncogenic driver in several subtypes of Diffuse Large B-Cell Lymphomas (DLBCLs)[28,29]. The BCL6 BTB domain interacts with co-repressors, such as SMRT and BCoR[30,31] and various independent academic and industrial groups have undertaken extensive efforts to develop small-molecules that inhibit BCL6 interactions or induce protein degradation[32,33]. The BCL6 BTB domain has proven to be a highly tractable target and, to date, many BCL6 inhibitors have been reported[34–39].

The BTB protein family contains numerous other members, several of which are potential targets for inhibitor development. For example, KAISO is a transcriptional repressor that recruits the co-repressor SMRT[40]. Depletion of KAISO has been reported to attenuate the survival of Triple-Negative Breast Cancer (TNBC) cells, suggesting that KAISO plays a role in TNBC oncogenesis[41]. LRF is another BTB-containing transcriptional regulator that recruits co-repressor complexes[42]. LRF regulates B-cell differentiation[43], and is also associated with the development of various cancers, including breast cancer[44] and prostate[45,46]. The BTB domain-containing protein MIZ1 functions in both the transcriptional activation and repression of target genes[47]. MIZ1 BTB domain interacts with HECT-type ubiquitin ligase HUWE1 (also called MULE)[48], and recent structural studies revealed an atypical binding mode with the dimeric BTB domain recognizing a single HUWE1 molecule[49]. MIZ1 also interacts with MYC through a motif outside of its BTB domain and has emerged as an oncogenic co-factor in the MYC-dependent medulloblastomas[50].

In this study, we evaluate the ligandability of BTB domains. BCL6, KAISO, LRF, and MIZ1 all possess structurally related and conserved BTB domains, yet small molecule inhibitors have been reported only for BCL6. Therefore, we explore whether the three other members of the BTB family are ligandable. Using fragment-based screening against the BTB domains of KAISO, LRF, and MIZ1, we identify multiple small-molecule ligands that bind to MIZ1. Surprisingly, we find no hits for KAISO and LRF. To rationalize this unexpected finding, and to elucidate the biophysical and structural bases of ligandability of these BTB family members, we investigate their dynamics using solution NMR spectroscopy. Rigorous analysis of spin relaxation data reveals that MIZ1 possesses a distinct dynamics profile compared to KAISO and LRF, featuring specific motions on the μs-ms time scale. Notably, the location of the MIZ1[BTB] residues with elevated dynamics coincides with the binding site of HUWE1[49] and the small molecule ligands we discover. We propose that protein dynamics represents a significant mechanism governing the recognition of small-molecule ligands by MIZ1[BTB]. Our data argue that protein dynamics may be a broadly applicable tool in drug discovery to assess the ligandability of novel and challenging targets.

## Results

### BTB domain of MIZ1 but not KAISO or LRF has a high propensity to bind small molecule ligands

To evaluate the respective ligandabilities of the LRF, KAISO, and MIZ1 BTB domains, we screened each protein against a library of 600 chemically diverse, fragment-like small molecules by protein-observed solution NMR spectroscopy. Specifically, recombinant, uniformly [15]N-labeled BTB domains of LRF, KAISO, and MIZ1, referred to herein as LRF[BTB], KAISO[BTB], and MIZ1[BTB], respectively, were screened with

mixtures containing 10 compounds via a series of [1]H-[15]N HSQC experiments. First, we screened MIZ1[BTB] and found that approximately 40 out of 60 screening mixtures yielded detectable chemical shift perturbations. To rank the hits according to their binding potencies, we calculated the sum of chemical shift perturbations of seven selected amide proton resonances (7PA value)[34]. Out of the 40 mixtures with 7 PA values > 100 Hz, we selected two mixtures yielding the most significant 7 PA values > 300 Hz for further analysis (Supplementary Fig. 1). We deconvoluted these mixtures and identified three compounds, 2GG4, 4CC2, and 5DD7 (Fig. 1a), that yielded the most pronounced chemical shift perturbations upon binding to MIZ1[BTB] (Fig. 1b). These hits comprise three chemically distinct scaffolds, which suggest that MIZ1[BTB] can bind structurally diverse compounds. To determine their respective binding affinities, we performed NMR-based titration experiments (Fig. 1b, c), and found that 2GG4 binds MIZ1[BTB] with the highest affinity ($K_d$ of $68 \pm 9$ μM, Fig. 1c), which is relatively potent for a small molecule fragment-like compound[51]. The two remaining compounds presented weaker affinities, with the $K_d$ values of $425 \pm 59$ and $870 \pm 140$ μM for 4CC2 and 5DD7, respectively (Fig. 1c). Next, we performed the fragment screen against KAISO[BTB] and LRF[BTB]. In contrast to MIZ1[BTB], we did not identify any compounds that bind to these proteins. These results emphasize remarkable differences in the propensity of three structurally related BTB domain proteins to bind small-molecule ligands: we found a 7% hit rate for MIZ1[BTB] (assuming one hit per mixture), whereas no hits were identified for KAISO[BTB] and LRF[BTB].

### Small molecule ligands bind to a conformationally variable site in MIZ1 BTB domain

The binding of 2GG4, 4CC2, and 5DD7 to MIZ1[BTB] leads to extensive chemical shift perturbations on the [1]H-[15]N HSQC spectra (Fig. 1b; Supplementary Fig. 1). To characterize the ligand-binding site, we mapped the chemical shift perturbations of methyl groups (Fig. 2a - upper row), as well as the backbone amides (Fig. 2a - bottom row), onto the structure of MIZ1[BTB]. These perturbations cluster around residues in strands B1, B2, and B4, and helices A2 and A3 (Supplementary Fig. 2 and Fig. 2a, b). Analysis of several independently determined crystal structures of MIZ1[BTB], including a structure determined in this study (Supplementary Table 1 and Supplementary Fig. 3), indicates that this region of BTB domain is conformationally variable. Residues 56–64, which comprise the B4 strand in a canonical BTB fold[52], namely β-strand, short α-helix, 3₁₀-helix, and loop, can adopt a variety of distinct conformations, or can be disordered (Supplementary Fig. 4). The MIZ1[BTB] domain thus exists in "closed" and multiple "open" conformations resulting from the structural variability around B4. This variable region is consistent with the small molecule binding site mapped by NMR (Fig. 2a, b). Analysis of NMR chemical shifts and through-space H[N]-H[N] NOE contacts revealed that residues 56–64 form a β-strand conformation with hydrogen bonds between B2 and B4 (Supplementary Fig. 5). Thus, the BTB domain adopts a predominantly a "closed" conformation in solution, which presents a compact structure. In contrast to the "open" form, the "closed" form lacks the pockets suitable for small molecule binding (Fig. 2a, b). We hypothesize that the compounds we identified bind to a transiently populated "open" conformation that features a pocket that is large enough to accommodate small molecules. Recent studies found that this site in MIZ1[BTB] is indeed involved in ligand binding and recognition of a peptide fragment from the HUWE1 E3 ligase[49]. The crystal structure of MIZ1[BTB] in complex with HUWE1 derived peptide demonstrates that BTB domain adopts an "open" conformation with the B2–B4 interface being involved in PPIs with HUWE1 (Fig. 2c). The MIZ1[BTB]-HUWE1 interface coincides with the binding site for the fragment compounds we identified (Fig. 2c) indicating that FBS is an unbiased method that can uncover PPI binding sites.

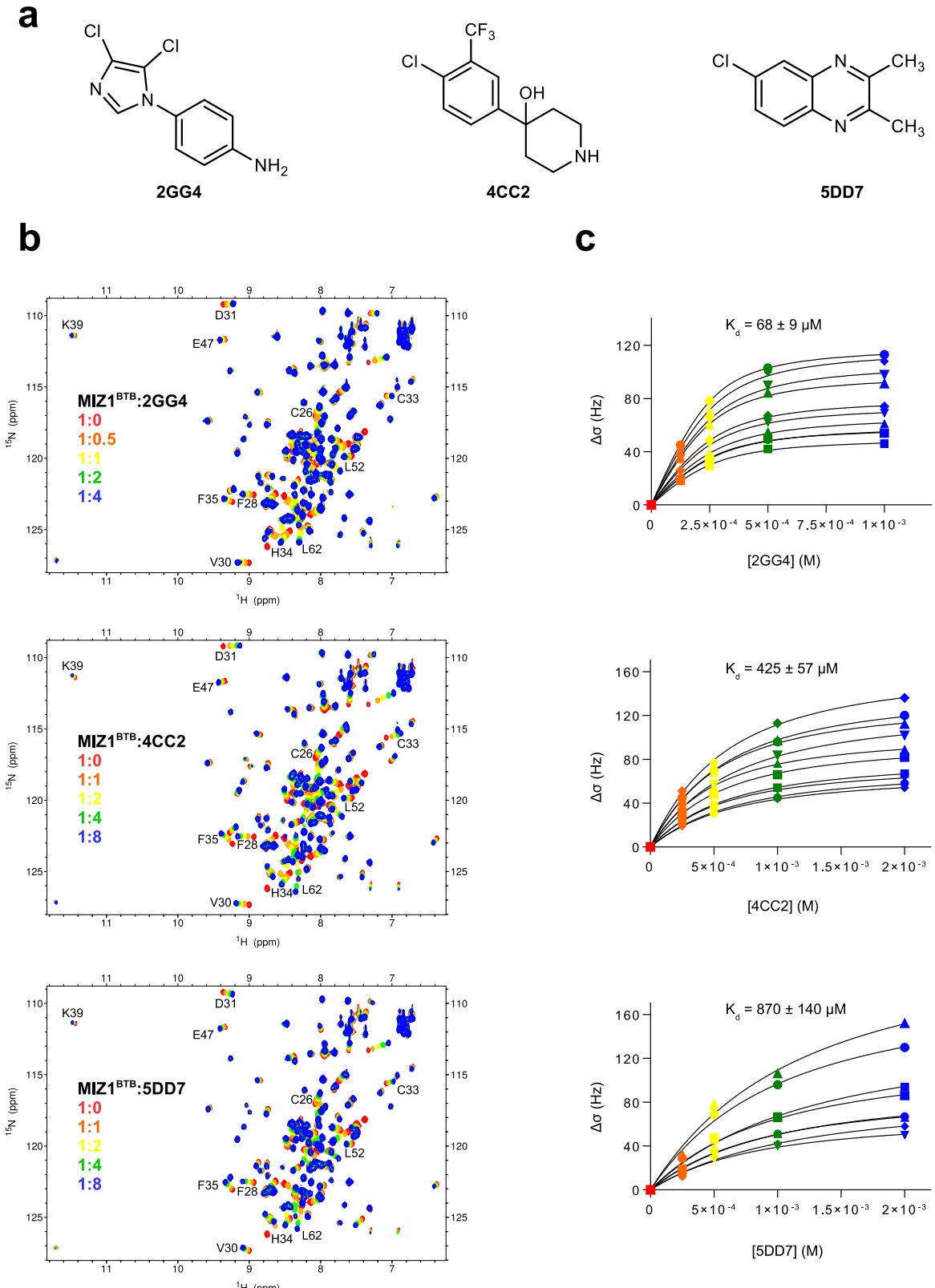

**Fig. 1 | MIZ1^BTB binds different classes of small molecule ligands. a** The chemical structures of fragments that bind to MIZ1^BTB; **b** The ¹H-¹⁵N HSQC spectra of MIZ1^BTB (red) titrated with the three fragments. The molar ratios of MIZ1^BTB-ligands are listed and colors correspond to coloring of the spectra. Selected residues experiencing large chemical shift perturbations are labeled. **c** Determination of $K_d$ values from NMR titration experiments for the three MIZ1^BTB ligands. Averaged binding constants are reported +/− SD and are calculated from fitting the titrations of several amides. Source data for (**c**) is provided as a Source Data file.

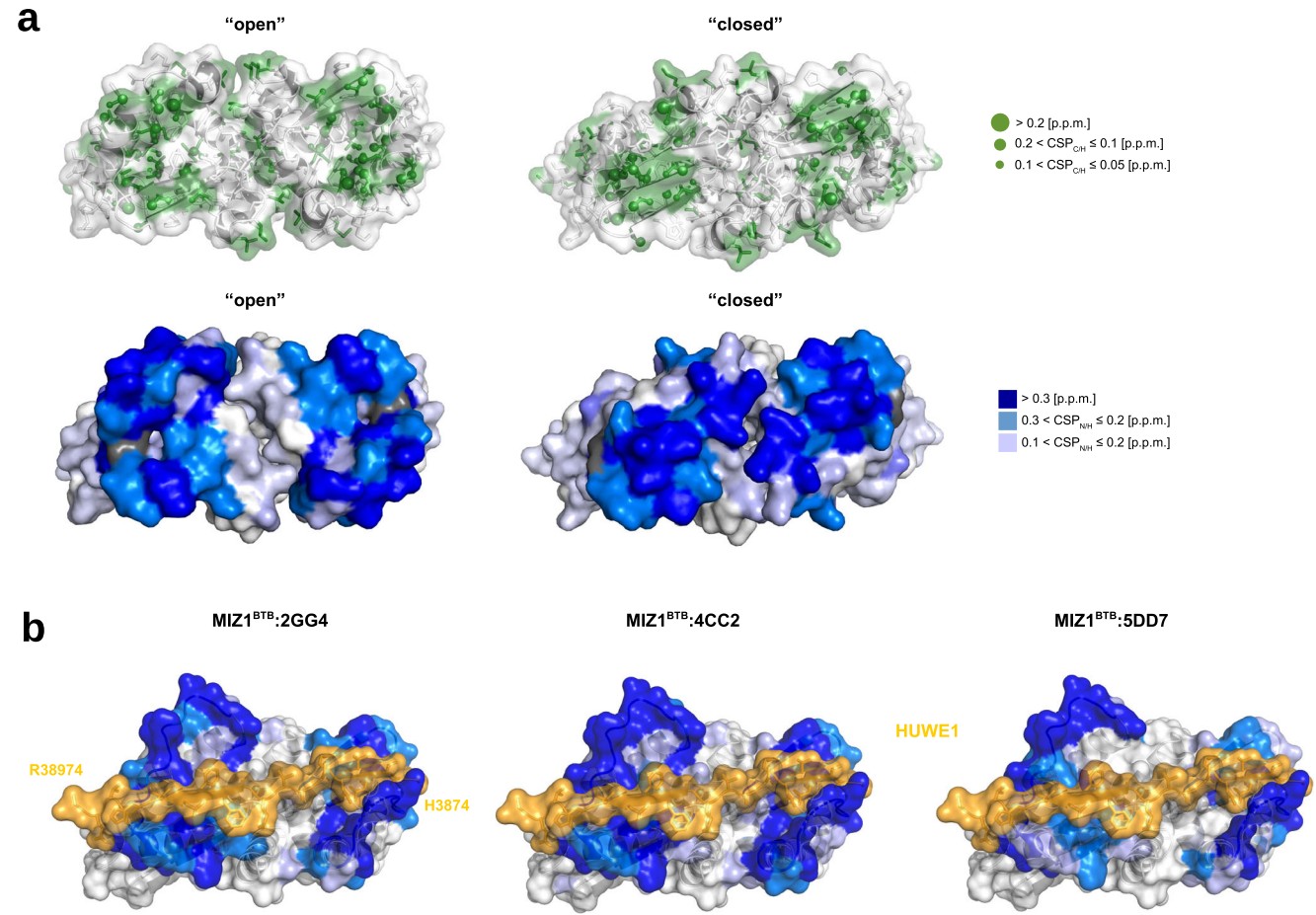

**Fig. 2 | Ligands bind to the conformationally variable site on BTB domain.**
**a** Mapping of the chemical shift perturbations of methyl groups (upper row) and backbone amides (lower row) resulting from the binding of 2GG4 to MIZ1[BTB]. The crystal structure is shown in "open" (PDB id: 2Q81:A) and "closed" (PDB id: 2Q81:B) conformations. All methyl-containing residues (ILVAMT) are shown in green with sphere radii corresponding to the magnitude of chemical shift perturbation.

Perturbations of backbone amides are in the dark-blue to light-blue colors corresponding to their $CSP_{N/H}$ effect. CSP effects correspond to the MIZ1[BTB]:2GG4 molar ratio of 1:4. **b** Mapping of the amide chemical shift perturbations $CSP_{N/H}$ upon binding of the three fragments onto the crystal structure of MIZ1[BTB] with bound HUWE1 (orange) (PDB id: 7AZX).

## MIZ1 BTB domain exhibits a dynamic profile distinct from KAISO and LRF

To assess whether ligandability of BTB domains can be predicted using computational methods, we employed FTMAP software, which can determine ligand binding hot spots based on protein structure[19]. First, we performed analysis of the BCL6[BTB] domain, which is highly ligandable, and we found multiple fragments mapped to a site encompassed by helices α2, α3, α6 and α1 from the second monomer (Supplementary Fig. 6). Importantly, this region represents a binding site for other known BCL6 inhibitors[34–39]. Then, we performed FTMAP analysis for LRF[BTB], KAISO[BTB], and MIZ1[BTB] and found several putative small molecule hot spots for all three BTB domains, including the same ligandable site as in BCL6[BTB] (Supplementary Fig. 6). Based on the FTMAP results we hypothesized that the propensity of MIZ1[BTB] to bind small-molecule ligands might result from distinct dynamics when compared to KAISO[BTB] and LRF[BTB]. To test this, we performed comprehensive studies of ¹⁵N relaxation for KAISO[BTB], LRF[BTB], and MIZ1[BTB] using two magnetic fields (Fig. 3; Supplementary Fig. 7). We found that all proteins are dimeric in solution with expected correlation times and size exclusion profiles (Supplementary Fig. 8). The comparison of the raw ¹⁵N spin relaxation observables reveals that all three BTB domains are well ordered, except for the flexible N- and C-termini. KAISO has a very rigid BTB domain, with the more flexible, short loop between B3-H4 (residues A64-G65) exhibiting elevated ps-ns motions, as indicated by decreased transverse relaxation rates accompanied by lowered NOE

values for those residues (Supplementary Fig. 7). LRF[BTB] has a similar dynamics profile to KAISO[BTB], with a longer, more flexible loop between B3 and H4 (residues S64-Q70) (Supplementary Fig. 7). Importantly, residues in both proteins show flat spectral density function ratio profiles at zero frequency, so-called Residue Consistency, i.e., $RC_i$ ratios at 1 (Fig. 3). These flat profiles indicate a lack of significant chemical exchange contribution $R_{ex}$[53]. Although several regions in KAISO[BTB] and LRF[BTB] present fast ps-ns motions, analysis of $RC_i$ values unequivocally indicates no contribution from dynamics on slower μs-ms time scales. In contrast, ¹⁵N relaxation parameters determined for MIZ1[BTB] reveal a distinct dynamics profile, particularly for B2 and B4 strands (Fig. 3). Residues in these regions of MIZ1 have $RC_i$ values that are significantly above 1, indicating the presence of elevated dynamics and slow motions on the μs-ms time scale (Fig. 3). Notably, the dynamics profile in MIZ1[BTB] clearly distinguishes this protein from KAISO and LRF, and correlates with its ability to bind small molecule ligands.

## MIZ1 small molecule ligand binding site shows enhanced μs-ms time scale dynamics

We found that small molecule ligands bind to the MIZ1 B2 and B4 strands (Fig. 2a, b), which show μs-ms time scale dynamics in ¹⁵N relaxation studies (Fig. 3; Supplementary Fig. 7). Thus, specific small molecule ligands bind MIZ1[BTB] at a site with elevated dynamics. To further quantify BTB domain dynamics, we performed a complete

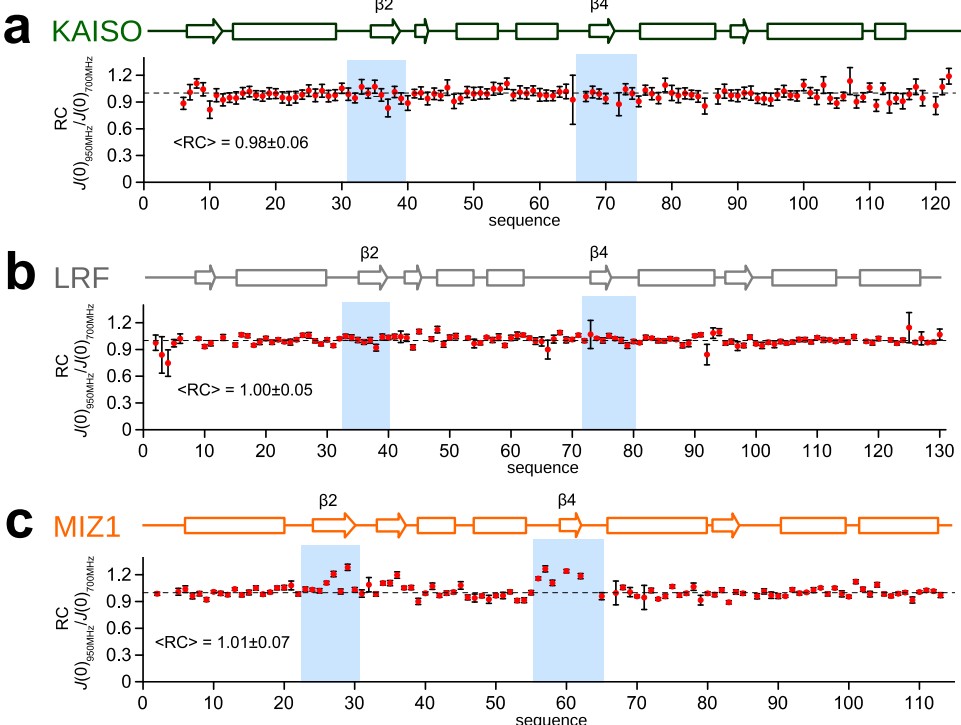

**Fig. 3 | BTB domains of KAISO, MIZ1, and LRF display distinct dynamic profiles.** The spectral density analysis plots derived from a comprehensive set of $^{15}$N spin relaxation observables, at two magnetic field strengths, for KAISO$^{BTB}$ (**a**), LRF$^{BTB}$ (**b**), and MIZ1$^{BTB}$ (**c**). The RC$_i$ ratio errors are propagated from the spin relaxation observables' uncertainties. Shaded regions indicate the isostructural positions of β4 and β2 strands, that only in MIZ1$^{BTB}$ are experiencing μs-ms time scale dynamics. Source data is provided as a Source Data file.

analysis of $^{15}$N relaxation data from two magnetic fields in the frame of the model-free approach (MFA)[54]. The MFA allows analysis of fast ps-ns local dynamics by the order parameter S$^2$, which characterizes the amplitude of motion and its internal time - $\tau_{int}$. MFA further detects slow μs-ms time scale collective motions, expressed as chemical exchange contribution, $R_{ex}$. Our MFA analysis confirms that the structurally conserved cores of KAISO$^{BTB}$ and LRF$^{BTB}$ are uniformly well ordered (average $S^2 = 0.94 \pm 0.02$), except for the flexible N- and C-termini (Fig. 4a, b). Additionally, the loop linking B3 and B4 in KAISO$^{BTB}$ experiences medium amplitude ps-ns motions (average $S^2 = 0.72 \pm 0.09$) (Fig. 4a, b), and is disordered in LRF$^{BTB}$ (average $S^2 = 0.51 \pm 0.07$) (Fig. 4a, b). However, in both proteins, B2 and B4 strands are well ordered. In contrast, MIZ1$^{BTB}$ displays elevated local dynamics for two regions encompassing B2 (average $S^2 = 0.84 \pm 0.03$) and B4 (average $S^2 = 0.74 \pm 0.08$) (Fig. 4a, b). Consistent with the analysis of the spectral density ratios (RC$_i$), the B2 and particularly B4 regions in MIZ1$^{BTB}$ specifically experience a contribution of chemical exchange to transverse relaxation ($R_{ex}$), indicating μs-ms dynamics around these motifs (Fig. 4c). The MFA analysis demonstrated that the crucial feature that distinguishes MIZ1$^{BTB}$ from KAISO$^{BTB}$ and LRF$^{BTB}$ is the presence of elevated dynamics on the μs-ms time scale for a subset of residues. Data from the fragment screens strongly suggest that elevated dynamics contribute to MIZ1$^{BTB}$ ligandability.

### Binding to HUWE1 quenches μs-ms dynamics in MIZ1 BTB domain

To evaluate whether μs-ms time scale dynamics in MIZ1$^{BTB}$ is an intrinsic feature of this domain or it is rather related to ligand binding, we determined the dynamics profile of MIZ1$^{BTB}$ in complex with HUWE1 peptide. Binding of HUWE1 breaks the symmetry of BTB domain dimer and results in doubling of the amide resonances of NMR spectra (Supplementary Fig. 9). We then assigned backbone resonances of MIZ1$^{BTB}$ in a complex with HUWE1 and collected spin relaxation

observables at two magnetic fields (Supplementary Fig. 10). Overall, binding of HUWE1 is not affecting the fast ps-ns dynamics in MIZ1, and order parameters for MIZ1$^{BTB}$-HUWE1 complex are in line with apo MIZ1$^{BTB}$ (Supplementary Fig. 11). Interestingly, the RC$_i$ profile for MIZ1$^{BTB}$-HUWE1 complex is flat, particularly for the B4 region, indicating a loss of $R_{ex}$ for the majority of residues at the interface with HUWE1 (Fig. 5a, b, d and Supplementary Fig. 12). Therefore, binding of the HUWE1 ligand suppresses the chemical exchange in MIZ1$^{BTB}$ suggesting that μs-ms dynamics plays a crucial role in ligand binding.

## Discussion

Prediction of protein ligandability or druggability is of paramount importance for discovery of small-molecule inhibitors of new protein targets. However, despite significant progress in the development of computational and experimental methods, inhibitor development remains challenging for many targets. Here, we uncovered striking differences in the ligandability of structurally related members of the BTB domain protein family. Our rigorous analysis indicates that ligandability of BTB domains correlates with the presence of μs-ms time scale dynamics.

BTBs are conserved domains, abundant in transcriptional regulators, and involved in the formation of protein–protein complexes. Since these proteins are associated with a range of diseases[27], BTB domains represent attractive targets for inhibitor development. To date, BTB domain inhibitors have been reported only for BCL6[33], raising the question of whether other BTB family members are ligandable. In this study, we performed fragment screens against three previously unexplored BTB domains of KAISO, LRF, and MIZ1. Surprisingly, we discovered that only the screen against MIZ1$^{BTB}$ yielded small molecule ligand hits, in stark contrast with the screens against KAISO$^{BTB}$ and LRF$^{BTB}$. We, therefore, examined why domains with conserved three-dimensional structures have such distinct propensities to bind small-molecule ligands, and tested what defines the

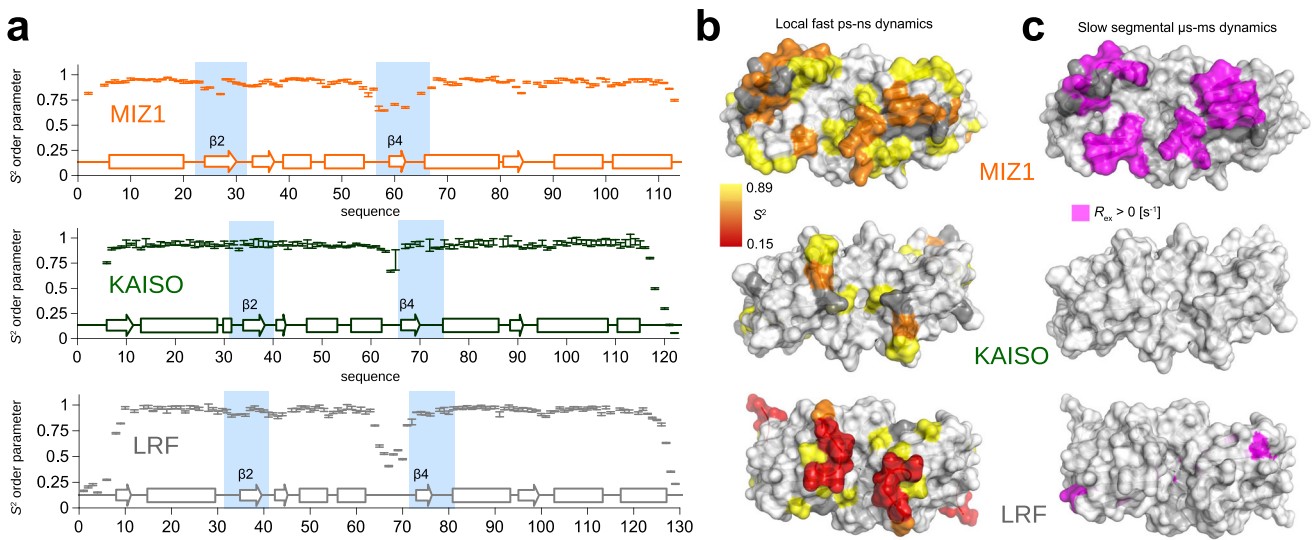

**Fig. 4 | MIZ1ᴮᵀᴮ experiences elevated µs·ms dynamics. a** Order parameters $S^2$ reporting on the local fast dynamics plotted along the MIZ1ᴮᵀᴮ (orange), LRFᴮᵀᴮ (gray), and KAISOᴮᵀᴮ (green) primary sequence. Values and corresponding errors are mean +/− SD as error bars obtained from propagated uncertainty after Monte-Carlo simulations. The secondary motifs are marked along the primary sequence and light-blue squares indicate B2 and B4 motifs. Source data is provided as a Source Data file. **b** The $S^2$ order parameters plotted on the structures of BTB domains scaled yellow to red to indicate increasing fast local dynamics. **c** Surface representations of the studied BTB domains with mapped residues with identified chemical exchange, $R_{ex}$ contribution, indicating the slow segmental µs·ms dynamics.

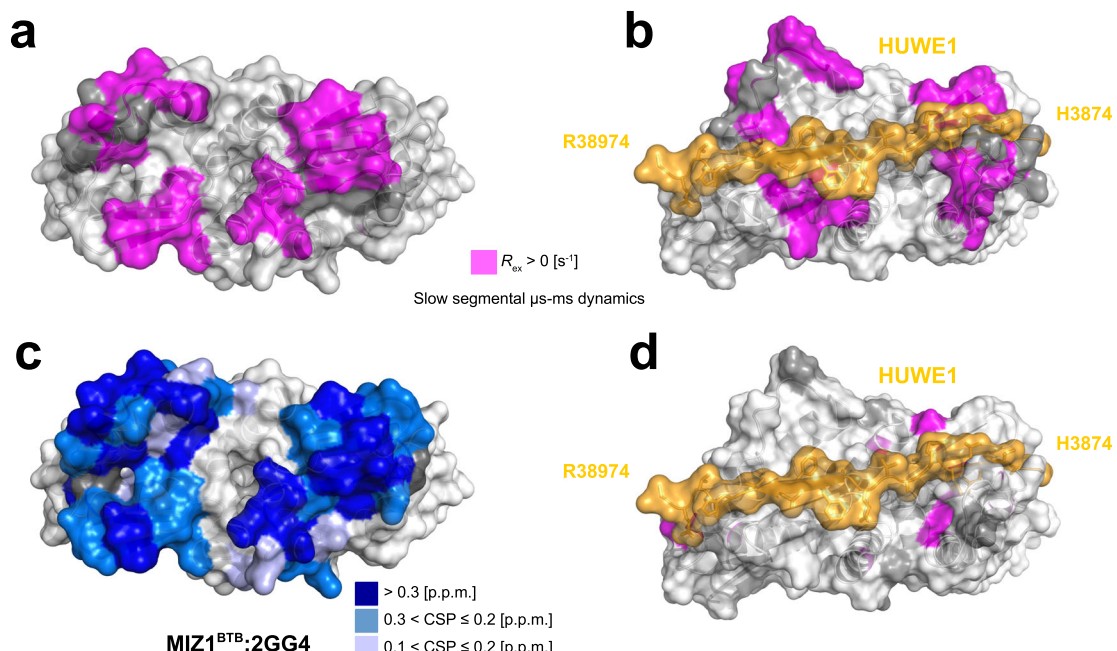

**Fig. 5 | Functional dynamics at µs·ms time scale coincides with the ligand binding site in MIZ1ᴮᵀᴮ. a, b** Mapping of residues that show slow µs·ms segmental motions indicated by the chemical exchange $R_{ex}$ (magenta), determined for the MIZ1ᴮᵀᴮ, onto the structure of the MIZ1ᴮᵀᴮ dimer (PDB id: 2Q81, panel **a**) and the structure of MIZ1ᴮᵀᴮ with bound HUWE1-derived peptide (PDB id: 7AZX; peptide is in orange, panel **b**); **c** backbone amide chemical shift perturbations upon 2GG4 binding mapped onto the structure of the MIZ1ᴮᵀᴮ dimer (PDB id: 2Q81); **d** the chemical exchange $R_{ex}$ (magenta) determined for MIZ1ᴮᵀᴮ bound to HUWE1 peptide mapped on the structure of MIZ1ᴮᵀᴮ-HUWE1 complex (PDB id: 7AZX).

ligandability of BTB domains. Computational prediction of ligand binding hot-spots solely based on the three-dimensional structures did not show any preference for ligandability among analyzed BTB domains, including the highly ligandable BCL6 (Supplementary Fig. 6). Thus, we used NMR spectroscopy to compare protein dynamics in solution. On the bases of experimentally determined order parameters $S^2$, we demonstrate that the cores of the three BTB domains are relatively rigid ($<S^2> = 0.93$), except for disordered N- and C-termini, and the flexible internal loop in LRF encompassing residues S64-Q70. We found that MIZ1ᴮᵀᴮ has a uniquely elevated dynamics profile on the slow µs·ms time scale in B2 and, particularly, in B4 region, which is indicated by significant contributions from $R_{ex}$ (Fig. 5a, b). The same B4 region experiences elevated fast local dynamics as evidenced by the average $S^2$ ranging from 0.74 to 0.84 (Fig. 4a, b). Corresponding

regions in KAISO and LRF do not show such µs-ms dynamics, and moreover, show local rigidity, with an average $S^2$ of 0.93 and 0.92 for KAISO and LRF, respectively. The KAISO and LRF BTB domains lack significant contribution from $R_{ex}$ (Fig. 4). Importantly, regions with elevated dynamics and $R_{ex}$ in MIZ1[BTB] correspond to the binding site for small-molecule ligands identified through FBS (Fig. 5c). We also found that binding of HUWE1 ligand does not affect fast local ps-ns motions in MIZ1[BTB] (Supplementary Fig. 11), in contrast, it quenches slow µs-ms motions (Fig. 5d). We determined that in solution, the interface of the free BTB domain involved in HUWE1 binding is different and adopts predominantly "closed" conformation, not accessible to ligand binding. However, µs-ms dynamics in concert with conformational heterogeneity observed in MIZ1[BTB] crystal structures (Supplementary Fig. 4) may reflect the presence of cryptic pockets suitable for ligand binding, and suggest that binding of HUWE1 might selects one of the conformations already present in MIZ1[BTB]. Overall, this example highlights the important role of µs-ms dynamics, which could be explored for ligand design (Supplementary Fig. 13).

BCL6 is an example of a highly tractable member of the BTB family, as demonstrated by multiple fragment hits identified in FBS campaigns[34,35], and by several reports of potent inhibitors[36–39]. We excluded BCL6[BTB] from the studies reported here, as wild-type BCL6[BTB] shows low solubility and requires several point mutations to obtain recombinant protein amenable to biophysical and structural characterization by NMR[34,55]. We anticipated that such mutations might perturb the protein's native dynamics pattern. Nevertheless, independent studies by our group[34] and by others[56] have found that ¹H-¹⁵N HSQC spectra of BCL6[BTB] feature peak broadening and lacking resonances, suggesting the presence of slow segmental µs-ms dynamics. Increased dynamics in the BCL6[BTB] at the interface with co-repressor peptide has additionally been observed in MD simulations[57]. Therefore, it is likely that BCL6[BTB] undergoes elevated dynamics, comparable to the dynamics measured here for MIZ1, which facilitates ligand binding.

BCL6 recruits binding partners in the lateral groove, formed at the interface of the BTB domain dimer[30,31]. The lateral groove on BCL6[BTB] also constitutes the binding site for all reported fragment-like compounds and small molecule BCL6 inhibitors[34–37,39]. Comparison of the crystal structure of BCL6[BTB]-SMRT co-repressor complex, with the crystal structures of KAISO[BTB], LRF[BTB], and MIZ1[BTB], confirmed that the putative lateral groove is present in all these BTB domains, and have been identified via FTMAP as potential small molecule binding sites (Supplementary Fig. 6). Our fragment screening campaigns failed to identify any small molecule ligands that bind to the lateral grooves on KAISO, LRF, and MIZ1, but did identify compounds binding to MIZ1[BTB] at a distinct site centered around B2, B4 strands. The recently reported crystal structure of MIZ1[BTB] in complex with a bound peptide fragment of HUWE1 revealed an unreported PPI binding site on MIZ1[BTB][49], which involves the residues in B2 and B4 strands and coincides with the binding site we identified through FBS. This again emphasizes that FBS is a robust approach to assess protein ligandability and identify functional binding sites in an unbiased manner. Here, we propose that the presence of slow dynamics in such sites may support the selection of proteins for the development of small molecule inhibitors.

Previous work established that small-molecule inhibitor of HUWE1 E3 ligase stabilizes MIZ1 and inhibits MYC-dependent transactivation in colorectal cancer cells[58]. Thus, disrupting MIZ1-HUWE1 interaction may prove to be viable strategy to develop new anti-cancer agents. To date, no inhibitors of MIZ1 have been reported and it is likely that the small molecule fragments we identified can be further developed into potent MIZ1[BTB] PPI interaction inhibitors.

In summary, we performed a comprehensive analysis that relates ligandability to the backbone dynamics of structurally related BTB domains. Based on these studies, we found that protein dynamics, in addition to structural and topological features, governs the ability of

BTB domains to bind small-molecule ligands. We anticipate that elevated µs-ms dynamics may represent a general feature of druggable proteins, and comprehensive studies of protein dynamics employing NMR relaxation may reveal novel, highly tractable drug discovery targets.

## Methods

### Expression and purification of BTB domains (LRF[BTB], MIZ1[BTB], KAISO[BTB])

Synthetic genes encoding *Human* BTB domains KAISO (residues 1–122), LRF (residues 1–130), and MIZ1 (residues 1–115) were ordered from Life Technologies and sub-cloned using *BamHI* and *HindIII* restriction sites into pet32a expression plasmid containing N-terminal Thioredoxin (Trx) and hexa-histidine (his₆) tags with an N-terminal PreScission-cleavage site. Recombinant pet32a-BTB domain plasmids were transformed into *E. coli* strain BL21 (DE3) cells and transformations were grown in Luria Broth (LB) or ¹⁵N-labeled M9 media with Ampicillin selection. Protein was expressed by induction with 0.25 mM IPTG for 16 h at 18 °C. Cells were lysed in buffer containing 50 mM Tris (pH 7.5), 300 mM NaCl, 30 mM imidazole, 1 mM TCEP, and 0.5 mM PMSF, and the clarified lysate was applied to Ni-NTA resin. Protein was eluted using lysis buffer with 300 mM imidazole, and Trx-his₆-tags were proteolytically cleaved with PreScission enzyme. Eluate was re-applied to Ni-NTA resin to extract Trx-his₆-tags, and KAISO[BTB] and MIZ1[BTB] were dialyzed extensively against 50 mM Tris (pH 7.5), 150 mM NaCl, and 1 mM TCEP at 4 °C for protein-observed solution-NMR experiments. For LRF[BTB], protein was dialyzed against 50 mM HEPES (pH 7.5), 150 mM NaCl, and 1 mM TCEP. For MIZ1[BTB] crystallization experiments, protein was dialyzed against 20 mM Tris (pH 7.5), 300 mM NaCl, and 1 mM TCEP. For the NMR resonance assignment, titrations with ligands and ¹⁵N spin relaxation studies the uniformly labeled U-¹⁵N and U-¹³C,¹⁵N BTB domain samples were obtained from M9 media by using 1 g/L of ¹⁵NH₄Cl as a sole source of nitrogen and 2.5 g/L ¹³C-glucose as a sole source of carbon. In order to obtain the perdeuterated U-²H,¹³C,¹⁵N KAISO[BTB] sample, the M9 media was prepared in 99.9% D₂O (CIL) with 2.5 g/L of ²H,¹³C-glucose and 1 g/L of ¹⁵NH₄Cl. The synthetic HUWE1³⁸⁷⁰⁻³⁸⁹⁷ peptide was ordered from GeneScript with >99% purity, TFA removal service and dialyzed to the buffer containing 50 mM HEPES (pH 7.5), 150 mM NaCl, 1 mM TCEP before the further use.

### Fragment screening by protein-observed solution-NMR spectroscopy

150 µM ¹⁵N-labeled KAISO[BTB] and MIZ1[BTB] in buffer comprised of 50 mM Tris (pH 7.5), 150 mM NaCl, 1 mM TCEP, and 7.5% D₂O was prepared for screening. LRF[BTB] was prepared in 50 mM HEPES (pH 7.5), but protein concentration and buffer conditions were otherwise maintained for KAISO[BTB] and MIZ1[BTB] screening. The library used for screening was comprised of commercially available fragment-like small molecules, and in-house synthesized compounds. Protein was incubated with fragment-like small molecules in mixtures of 10 compounds per sample, at 500 µM per compound, and 5% DMSO (*v/v*). ¹H-¹⁵N HSQC spectra were acquired at 30 °C on 600 MHz Bruker Avance III Spectrometer equipped with cryogenic probe running Topspin version 3.5. Processing and visualization of HSQC spectra were performed in NMRPipe ver. 11.0 and Sparky ver. 3.314. Fragment hits were identified by chemical shift perturbations on ¹H-¹⁵N HSQC spectra. The chemical shift perturbation of ¹H/¹⁵N resonances was determined with the equation (1): $CSP_i = [(\Delta\delta_{H,i})^2 + 0.1 \cdot (\Delta\delta_{N,i})^2]^{0.5}$, where $\Delta\delta_{H,i}$ and $\Delta\delta_{N,i}$ are detected chemical shift changes of proton and nitrogen, respectively.

### $K_D$ determination by protein-observed solution-NMR spectroscopy

¹H-¹⁵N HSQC spectra were obtained using samples with 250 µM ¹³C,¹⁵N-labeled MIZ1[BTB] in buffer composed of 50 mM Tris (pH 7.5),

150 mM NaCl, 1 mM TCEP, 7.5% $D_2O$, and 5% DMSO. NMR-titration experiments were performed using increasing concentrations of compound 2GG4 (125, 250, 500, and 1000 μM), and of compounds 4CC2 and 5DD7 (250, 500, 1000, and 2000 μM). Dissociation constants were derived by least-squares fitting of chemical shift perturbations as function of ligand concentration (2):

$$\delta_i = \{b - \sqrt{(b^2 - 4 \times a \times c)}\}/2a \qquad (2)$$

with $a = (K_a/\delta_b) \times [P_t]$, $b = 1 + K_a([L_{ti}] + [P_t])$, and $c = \delta_b \times K_a \times [L_{ti}]$, where $\delta_i$ is the absolute change in chemical shift for each titration point, $[L_{ti}]$ is the total ligand concentration at each titration point, $[P_t]$ is the total protein concentration, $K_a = 1/K_d$ is the binding constant, and $\delta_b$ is the chemical shift of the resonance in question in the complex. $K_d$ and $\delta_b$ were used as fitting parameters in the analysis.

### Determination of MIZ1[BTB] crystal structures

Screening by the hanging-drop vapor-diffusion technique was used to first obtain crystals of MIZ1[BTB]. Crystals were then optimized using the sitting-drop technique, via incubation of equal volumes (1.5 μL) of protein [6 mg/mL in 20 mM Tris (pH 7.5), 300 mM NaCl, and 1 mM TCEP] and crystallant solution (30% PEG-1500 [*w/v*]). Crystals formed over one week at 4 °C. Prior to freezing, crystals were transferred to cryoprotectant solution comprised of crystallant solution with 25% glycerol.

### Crystallographic data collection and structure determination

Diffraction data for MIZ1[BTB] was collected at 21-ID-D beamlines at the Life Sciences Collaborative Access Team at the Advanced Photon Source. Data were integrated and scaled using HKL-2000, and structures were solved by molecular replacement using known native MIZ1[BTB] structure as the search model. The model was refined using REFMAC, COOT, CCP4 program suite, and PHENIX program suite. Structure validation was performed using MOLPROBITY and ADIT servers. Data collection and structure refinement statistics are reported in Supplementary Table 1.

### Sequence-specific resonance assignment of globular BTB domains

The backbone $^1H^N$, $^{15}N$, $C_\alpha$, $C_\beta$ and C chemical shifts of LRF[BTB], MIZ1[BTB], KAISO[BTB] were assigned based on a set of TROSY-type 3D triple resonance experiments, i.e., HNCA, HNcoCA, HNcoCACB, HNCACB, CBCAcoNH, HNCO, HNcaCO and HNCB supported by 3D $^{15}N$-edited NOESY-HMQC experiment (mixing times 100 ms for U-$^{13}C$,$^{15}N$-labeled samples and 200 ms for perdeuterated U-$^2H$,$^{13}C$,$^{15}N$ samples)[59,60]. In contrast to LRF[BTB] and MIZ1[BTB], where a decent quality of 3D triple resonance spectra was obtained from uniformly double labeled U-$^{13}C$,$^{15}N$-samples at concentrations of 90–160 μM (per dimer), the KAISO[BTB] domain and MIZ1[BTB]:HUWE1 complex required perdeuteration. Analogical 3D triple resonance TROSY-type experiments were recorded utilizing $^2H$-decoupling on 100 μM (per dimer) samples of U-$^2H$,$^{13}C$,$^{15}N$ KAISO[BTB] and U-$^2H$,$^{13}C$,$^{15}N$ MIZ1[BTB] with unlabeled HUWE1$^{3870-3897}$ peptide (1:3 molar ratio). The following buffer conditions were used for all samples: 50 mM HEPES (pH 7.5), 150 mM NaCl, 1 mM TCEP, and 1% (*v/v*) $D_2O$. All spectra were recorded at 30 °C on 700 and 950 MHz Bruker Avance NEO spectrometers equipped with 5 mm TCI cryogenic probes. Spectra were processed in TOPSPIN 4.0.7 and analyzed in CARA ver. 1.8 software. The secondary structure motif propensities for structures in solution were determined with Talos-N software from backbone $^1H^N$, $^{15}N$, $^{13}C_\alpha$, $^{13}C_\beta$, and $^{13}C$ chemical shifts[61].

### Complete methyl group assignment of MIZ1[BTB] and titrations with ligands

The complete set of $^1H$/$^{13}C$ resonances from the methyl moiety containing amino acids, i.e., alanine, threonine, leucine, isoleucine, valine and methionines, was achieved for uniformly labeled $^{13}C$,$^{15}N$-MIZ1[BTB] sample of 200 μM (per dimer). The side-chain methyl assignment was started from the backbone resonance assignments and a combination of high-resolution through-bond correlations in 3D spectra recorded on 950 MHz instrument: 3D (H)CCH-TOCSY (mixing times of 5.6 ms giving nearly COSY-type correlation mainly through three bonds, and 16.4 ms for total through multiple C−C bonds correlations). Moreover, the through-space correlations from 3D $^{15}N$-edited NOESY-HMQC (mixing time 100 ms) and 3D $^{13}C$-edited NOESY-HSQC (mixing time 100 and 200 ms) centered on the aliphatic region with $^1J_{C-H}$ set to 125 Hz were used together with available static X-ray structures from PDB to verify the assignments. The hydrogens were added in Coot ver. 0.8.9.2 software. The methionine methyl groups missing the TOCSY-type correlations were identified based on the negative sign on 13.3 ms constant time 2D $^1H$-$^{13}C$ CT-HSQC spectra and their methyls correlated to their amide via NOE cross peaks present on $^{13}C$- and $^{15}N$-edited NOESY spectra. Two 2D $^1H$-$^{13}C$ HSQC constant time (CT time of 13.3 with Methionine -S-$CH_3$ and -$CH_2$- groups negative and 26.6 ms with all correlations positive) at high-resolution allowed to unequivocally resolve methyl signals. The NMR titrations of $^{13}C$,$^{15}N$ MIZ1[BTB] with selected ligands were performed as described above and monitored with 2D $^1H$-$^{13}C$ CT-HSQC spectra centered on the methyl group region. Spectra were processed in TOPSPIN 4.0.7 and analyzed in CARA software.

### Relaxation measurements and BTB domain dynamics analysis

The $^{15}N$ backbone amide spin relaxation experiments were collected at 700 and 950 MHz ($^1H$ frequency) instruments on uniformly labeled U-$^{15}N$ samples of all three BTB domains and the MIZ1[BTB]:HUWE1 complex at an identical concentration of 100 μM each with 50 mM HEPES (pH 7.5), 150 mM NaCl, 1 mM TCEP, and 1% (*v/v*) $D_2O$. The NMR experimental temperature was carefully calibrated to 30 °C with standard ethylene glycol (80%) in DMSO-$d_6$ sample at two magnetic fields operating at $^1H$ frequencies of 700 and 950 MHz. The temperature was monitored before and after each experiment and found to be stable within ±0.3 °C. The upgraded TROSY-versions of $^{15}N$-$R_1$, $^{15}N$-$R_2$, and {$^1H$}-$^{15}N$ heteronuclear NOE experiments optimized for larger molecular weight proteins were utilized[62,63]. The recycle delay of 4 s for $^{15}N$-$R_1$ and $^{15}N$-$R_2$ and 10 s for {$^1H$}-$^{15}N$ NOE (10 s of the $^1H$-saturation period) were used. The sixteen delays from 0 to 3520 ms were used for $^{15}N$-$R_1$ and nineteen from 0 to 250.16 ms for $^{15}N$-$R_2$, for both cases in a randomized order. The relaxation rates were fit from the intensities to exponential decay curves with two parameters in a non-linear least-square procedure, and standard errors were derived from variance-covariance matrix analysis using QtiPlot ver. 0.9. The {$^1H$}-$^{15}N$ heteronuclear NOE values were obtained from the ratio of intensities and errors from the propagation of the signal-to-noise values of two signals corresponding to an individual residue. Residues with the complete set of $^{15}N$-$R_1$, $^{15}N$-$R_2$, and {$^1H$}-$^{15}N$ NOE parameters from two fields were subjected to model-free analysis (MFA). The dynamics interpretation based on the collected $^{15}N$ spin relaxation data was performed according to our previous approaches[54,64,65]. In short: a fully anisotropic model of motion was applied for all proteins, with the assumption of C2 symmetry for each homo-dimer resulting in four global parameters, $D_1$, $D_2$, $D_3$ and $\gamma$ Euler angle (while $\alpha = \beta = 0^\circ$). Local parameters of order parameter $S^2$ reporting on the amplitude of motion [0,1] and local $\tau_{int}$ time characterizing the ps-ns fast motions, and $R_{ex}$ term sensitive to slow μs-ms time scale motions, being the addition to the transverse $^{15}N$-$R_2$ relaxation rates. The coordinates were taken from the corresponding PDB files (KAISO[BTB] 3M4T; MIZ1[BTB] 2Q81:B symmetrized to reflect the major form in solution; LRF[BTB] 2NN2; MIZ1[BTB]:HUWE1 7AZX; the missing heavy and H-atoms were added, and energy-minimized in Coot ver. 0.8.9.2 software). Errors were obtained after Monte-Carlo procedure after 200 minimizations.

## Size exclusion chromatography with multi-angle static light scattering (SEC-MALS)

The SEC-MALS measurements were performed with MIZ1[BTB], KAISO[BTB], LRF[BTB] domains at 100 µM passed through the Superdex 200 30/300 column (GE Healthcare) using Agilent HPLC following DAWN® MALS detector. The running buffer contained 50 mM HEPES (pH 7.5), 150 mM NaCl, and 1 mM TCEP. Data were analyzed with the ASTRA® software provided by the company (Wyatt Technologies). The presented results are mean values with standard error from the three sample replicates.

## Data availability

The experimental crystallographic data and structural coordinates of the MIZ1 BTB domain determined in this study were deposited in the Protein Data Bank (PDB) under the accession code: PDB id 7T58. The following PDB data sets were used in the study: 2Q81, 1R29, 2NN2, 3M8V, 7AZX, 3M4T. Source data are provided with this paper.

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

## Acknowledgements

This publication is based upon work supported by the King Abdullah University of Science and Technology (KAUST) Office of Sponsored Research (OSR) under Award No. OSR-CRG2018-3792 and Award No. OSR-CRG2019-4088 (L.J.). This work was funded by the National Institute of Health (NIH) R01 grant CA207272 (T.C.) and T.C. is a Rogel Scholar. This research used resources of the Advanced Photon Source, a U.S. Department of Energy (DOE) Office of Science User Facility operated for the DOE Office of Science by Argonne National Laboratory under Contract No. DE-AC02-06CH11357. Use of the LS-CAT Sector 21 was supported by the Michigan Economic Development Corporation and the Michigan Technology Tri-Corridor (Grant 085P1000817).

## Author contributions

L.J. and T.C. were responsible for initiating and supervising the project and writing the manuscript. V.K., B.L., M.B., and I.Cz. prepared the proteins and performed biophysical characterization. V.K., M.J., J.G., L.J., and T.C. performed the structural biology, the NMR studies and dynamics studies. All authors contributed to data analysis and writing of the manuscript.

## Competing interests

T.C. and J.G. received prior research support from Kura Oncology Inc. for an unrelated project, served as consultants for Kura Oncology, and have equity ownership in the company. The remaining authors declare no competing interests.
