## [Peer Review File · Nature Communications]

REVIEWER COMMENTS

Reviewer #1 (Remarks to the Author):

The MS by Kharchenko et al presents the results from an NMR-based fragment screen on the BTB domains from Miz1, LRF and Kaiso. They obtained several hits for Miz1 but not for LRF or Kaiso. Next, they carried out a series of NMR experiments to demonstrate that three of the Miz1 hits bind in a dynamic/flexible part of the protein. The Kaiso and LRF BTB domains do not have regions of flexibility. They also generate a crystal structure of unliganded Miz1 BTB.

1. While the experimental work is sound, the interpretation and conclusions are problematic. Based on the correlation of the number of FBS hits with dynamics on their three targets (Miz1, Kaiso and LRF), they conclude that dynamic regions of proteins are more likely sites for ligand binding in BTB domains and other challenging targets. They base this conclusion on three data points - which makes for a very weak case of correlation. They go as far as to state:

“This finding suggests that protein dynamics, but not structural features, govern the ligandability of BTB domains.” (top of p. 15).

The principles of conformational selection and induced fit have been discussed by Koshland in 1960 and countless others since then. These principles are textbook biochemistry and few would disagree that protein dynamics are often (but not always) an important feature of ligand binding sites. Thus, the main point of the paper demonstrates an accepted fact. But to go further and say that structural features are **not** important in structure-based drug design would require a stronger body of evidence, to say the least.

2. They discuss BCL6, a related BTB domain that is highly ligandable, but omit the fact that the ligand-binding pocket is not any more dynamic than the rest of the protein. As a fourth data point, BCL6 does not fit their proposed correlation. Protein dynamics are often, but not always, involved in ligand binding.

3. The sequence identities of Miz1 BTB with Kaiso, LRF and BCL6 are 31%, 42% and 38%, respectively. The backbone structures are similar, but they are hardly “closely related” (p. 14).

4. Suppl. Fig. 6 shows models of a SMRT peptide bound to Kaiso, Miz1 and LRF. In fact, this BCL6-binding peptide does not bind to Miz1 (Stogios et al. *J Mol Biol* 400:983-97 (2010)) or LRF (Stogios et al. *Prot Sci* 16:336-42 (2007)), and probably not Kaiso.

5. The structural variability of Miz1 sheet B4 are well known and have been discussed in previously (papers associated with Suppl. Fig. 3). The extensive new NMR data presented here adds to our understanding of this region.

Overall, while the work presented here is of high technical quality, the findings are more

suited for a specialized audience.

Gil Privé

Reviewer #2 (Remarks to the Author):

Excellent paper - really nice piece of work, well thought out experiments to follow up an interesting observation. The correlation between backbone dynamics, FBS output and cryptic sites is a really interesting one, and will be of interest to the wider community.

My only negative comments are rather trivial - there is a typo on p8 ("trough space" rather than "through space") and on p21 the authors may mean "13.3ms constant time" rather than "contact time" which usually refers to ssNMR). Apart from these minor points, the manuscript is well written, well reasoned and clear.

Reviewer #3 (Remarks to the Author):

This manuscript by Kharchenko et al. presents an interesting investigation of the ligandability of a family of domains called BTB domains. The authors show that, out of three domains investigated, only one binds some fragments from a good-size library while the other two do not bind any fragment from this library. Interestingly, the domain that binds fragments is the only one that shows the presence of a conformational equilibrium as detected by chemical-exchange NMR. Strikingly, there is near perfect overlap between the region perturbed by ligand binding and the one undergoing conformational exchange. The authors conclude that such microsecond-millisecond dynamics can be a marker of ligandability. This is a very nice piece of work, which should be of interest to a wide audience of structural biologists, biophysicists, and all chemists involved in drug development. However, I would like to invite the authors to investigate a few major points before this manuscript is suitable for publication. I will list below both essential, major points, as well as minor changes.

Major points:

1. The authors show and discuss the excellent correlation between dynamics and ligand binding. Yet, I do not think that they manage to demonstrate causation. The sentence on page 15 "This finding suggests that protein dynamics, but not structural features, govern the ligandability of BTB domains" does not lay on solid enough ground. First, the authors present no physical model that could explain why dynamics leads to ligand binding. Second, if I may clearly state a hypothesis, which surprisingly appears in the abstract but not anywhere else in the manuscript: the most likely explanation is that the conformational exchange identified is between a major state unable to bind and a minor state that exposes a so-called cryptic pocket, that is much more prone to ligand binding. Determining the structure of such a minor state is difficult and only a few investigations so far have been able to determine such conformations. However, I can suggest a few investigations that could be done to probe this

hypothesis. For instance, if the minor state is similar to the bound conformation to HUWE1, the authors could compute the ligandability of the structures of MIZ1_BT B free and bound (without HUWE1 of course). Have the authors tried to co-crystallize MIZ1_BT B and 2GG4? This would definitely identify the putative cryptic pocket. The authors may also suggest other approaches. To conclude on this point, if the authors were to show that a cryptic pocket is only exposed in a minor conformational state in MIZ1_BT B, this would be a fantastic result, of interest to a very wide audience.

2. In the case of KAISO BT B, the elevated nitrogen-15 transverse relaxation rates (and the need for deuteration) point to multimers larger than dimers. In the case of KAISO but also all other BT B domains, could the authors investigate the multimers present in solution: are overall correlation times from the model-free analysis in agreement with the dimeric form for MIZ1 and LRF? Is KAISO_BT B in equilibrium between a dimer and a dimer of dimer? What are the exposed surfaces in dimers or larger multimers? These questions are essential when comparing ligandability.

Minor points:

- Could the authors discuss the solubility of the fragments presented in Figure 1?
- In Figures 2A-B and 2C, it is somehow difficult to compare the different views. Could the authors present the structure of the complex (2C) in the same orientation as panels 2A and 2B?
- With respect to fast motions, it is clear in Figure 4A that the beta4 strand is particularly dynamic in MIZ1. However, the picture is more contrasted for beta2, where only a handful of residues show slightly lower order parameters. This should be reflected in the way the authors discuss these data.
- Page 12, the authors write “a strong contribution from the chemical exchange (Rex)”, a more rigorous phrasing would be “a contribution of chemical exchange to transverse relaxation (Rex)”

Response to Reviewers' comments

We thank all Reviewers for constructive and valuable comments regarding our manuscript. Based on these comments, we further improved the manuscript and corrected some flaws and incorporated all suggested changes to the original submission. We have also added new experimental data to address Reviewer's comments. The most relevant changes strengthening the impact of our work are:

- We have performed computational analysis of the BTB domains from BCL6, MIZ1, LRF and KAISO using FTMAP software to evaluate ligandability based on the crystal structures of these proteins. We found that similar ligand binding hot-spots were identified for all four BTB domains, which is inconsistent with experimental fragment screening results, and further emphasizes the novelty of our findings, underlying the important role of protein dynamics in the prediction of ligandability. The description and the analysis of these results are included in the revised manuscript.
- To evaluate the role of the dynamics in the MIZ1-BTB on ligand binding, we have determined experimental dynamics for the MIZ1-HUWE1 complex. This required purification of [²H,¹³C,¹⁵N] MIZ1, the backbone resonance assignment and the extensive set of relaxation measurements at two magnetic fields and meticulous data analysis. Interestingly, we obtained very exciting results demonstrating that the ligand binding to MIZ1 quenches the slow dynamics on the μ s-ms time scale but not fast ns-ps dynamics. The HUWE1-binding driven suppression of the slow dynamics further indicates that μ s-ms time scale dynamics is relevant for ligand binding and is not just constituting an intrinsic property of MIZ1. Analysis of these new results further strengthens the manuscript.

We would like to point out that in a recent publication from *Nature Communications*, 2022, NMR relaxation studies were used to discover distinct flexibility in complexes of Mdmx and MDM2 with nutlin-3a (<https://www.nature.com/articles/s41467-022-28721-x>). Targeting flexible regions was employed to develop enhanced Mdm2/MdmX inhibitor with anti-cancer activity. This elegant example demonstrates that exploiting protein dynamics may support optimization of the potency of small molecule inhibitors. On the other hand, our studies emphasize the role of protein dynamics in identification of small molecule ligands at early stage of drug discovery process, which we believe is of broad interest to drug discovery community.

In summary, we have also addressed all other comments by performing additional experiments and revising the text. All changes to the text in the manuscript are marked in red. Detailed responses to Reviewers' comments are below.

Reviewer #1

1. While the **experimental work is sound**, the interpretation and conclusions are problematic. Based on the correlation of the number of FBS hits with dynamics on their three targets (Miz1, Kaiso and LRF), they conclude that dynamic regions of proteins are more likely sites for ligand binding in BTB domains and other challenging targets. They base this **conclusion on three data points** - which makes for a very weak case of correlation. They go as far as to state:

"This finding suggests that protein dynamics, but not structural features, govern the ligandability of BTB domains." (top of p. 15).

*The principles of conformational selection and induced fit have been discussed by Koshland in 1960 and countless others since then. These principles are textbook biochemistry and few would disagree that protein dynamics are often (but not always) an important feature of ligand binding sites. Thus, the main point of the **paper demonstrates an accepted fact**. But to go further and say that structural features are ***not* important in structure-based drug design** would require a stronger body of evidence, to say the least.*

Response:

We thank the Reviewer for noting this important flaw in our original manuscript. We fully agree that, without any doubt, the structural features of proteins are essential for small molecule binding and for that reason, scientists are solving structures of protein-inhibitor complexes. This statement has been corrected in the revised manuscript to:

"...protein dynamics, in addition to structural features, governs the ability of BTB domains to bind small-molecule ligands".

We are well aware of conformational selection and induced fit mechanisms proposed for ligand binding. However, the primary motivation for our studies was a surprising result from the fragment screening performed for LRF, KAISO and MIZ1, revealing a distinct propensity to bind small molecule ligands by the structurally similar BTB domains. To further investigate whether the crystal structures of these BTB domains may explain ligandability, we performed computational prediction of ligand binding hot spots using FTMAP software. Such analysis revealed that similar hot spots have been mapped to all three proteins and are similar to the highly ligandable BCL6 BTB domain (see Supplementary Fig 5). Furthermore, FTMAP prediction based on the "closed" structure of MIZ1 BTB consistent with the conformation in the solution (see Supplementary Fig. 4) did not identify hot spots in a site corresponding to the HUWE1 site on MIZ1 (see Supplementary Fig 12), which further exemplifies limits in prediction ligandability solely based on structural features. Based on our findings, we believe that prediction of ligandability based on protein dynamics of the BTB domains is novel and of interest to a broad interdisciplinary audience, and may support drug discovery efforts for this class of proteins.

To further assess whether dynamics plays a role in ligand binding or is an intrinsic feature of the MIZ1 BTB domain, we have performed NMR relaxation studies for the MIZ1-HUWE1 complex (these experiments proved very challenging due to loss of symmetry of BTB dimer in complex with HUWE1 and doubling of signals on NMR spectra). Interestingly, we found that binding of HUWE1 quenches the slow μ s-ms motions but not the fast ps-ns dynamics in MIZ1 BTB (Figure 5d, Supplementary Figures 9, 10, 11), again suggesting that the presence of slow dynamics is a feature governing ligand binding.

We also agree with the Reviewer that analysis performed for three proteins cannot be too broadly generalized. Nevertheless, we want to emphasize the unprecedented value of our study and note that the reliable measurements of protein relaxation and dynamics are time-demanding and require meticulous approach. We believe that our analysis correlating fragment screening and protein dynamics

is the first of this kind. Expanding such studies to other BTB domains or other proteins is a subject of ongoing research in our labs, and in our opinion, it is beyond the scope of this manuscript.

2. They discuss BCL6, a related BTB domain that is highly ligandable, but omit the fact that the ligand-binding pocket is not any more dynamic than the rest of the protein. As a fourth data point, BCL6 does not fit their proposed correlation. Protein dynamics are often, but not always, involved in ligand binding.

Response:

We agree with the Reviewer that the BCL6 BTB domain is highly ligandable. However, there is no experimental data in the literature about the dynamics of the BCL6 BTB domain. The majority of structural studies with BCL6 BTB, including NMR assignments, have been performed with protein mutant that contained mutations of three cysteine residues (C8Q, C67R and C84N) to enhance solubility. We have excluded the BCL6 BTB domain from our studies because such mutations may alter protein dynamics. Interestingly, the available NMR spectra of BCL6 BTB triple mutant show broadening of multiple resonances, and previous publications reported incomplete assignments for this protein (e.g. Cerchietti et al., *Cancer Cell*, 2010 Apr 13;17(4):400; Cardenas et al. *J Clin Invest*. 2016 Sep 1;126(9):3351), suggesting the presence of dynamics at a slower time scales.

Here, we would also like to emphasize that small molecule hot spot mapping using FTMAP software highlighted the lateral groove as a potential ligand binding site for BCL6 and also for the remaining three BTB domains (see Supplementary Fig. 5). This again emphasizes that structural features alone are not sufficient to distinguish or predict ligandability of BTB domains when employing computational techniques.

3. The sequence identities of Miz1 BTB with Kaiso, LRF and BCL6 are 31%, 42% and 38%, respectively. The backbone structures are similar, but they are hardly “closely related” (p. 14).

Response – we changed the statements in the manuscript stating that BTB domains of MIZ1, KAISO and LRF are structurally related, not closely related.

*4. Suppl. Fig. 6 shows models of a SMRT peptide bound to Kaiso, Miz1 and LRF. In fact, this BCL6-binding peptide does not bind to Miz1 (Stogios et al. *J Mol Biol* 400:983-97 (2010)) or LRF (Stogios et al. *Prot Sci* 16:336-42 (2007)), and probably not Kaiso.*

Response: we agree with the Reviewer that SMRT is not binding to the above-specified site in other BTB domain proteins, and the purpose of this figure is to show the presence of similar lateral grooves in the crystal structures of all three BTB domains. In fact, this site has been identified via computational FTMAP analysis as a possible small molecule binding pocket in four BTB domain proteins we evaluated (Supplementary figure 5).

We have revised the manuscript and added “putative lateral groove”, and we have included the following sentence:

"Comparison of the crystal structure of BCL6^{BTB}-SMRT co-repressor complex, with the crystal structures of KAISO^{BTB}, LRF^{BTB}, and MIZ1^{BTB}, confirmed that the putative lateral groove is present in all these BTB domains and that KAISO^{BTB}, LRF^{BTB}, and MIZ1^{BTB} all feature similar pockets (Supplementary Fig. 13), which have been identified via FTMAP as potential small molecule binding sites."

5. The structural variability of Miz1 sheet B4 are well known and have been discussed in previously (papers associated with Suppl. Fig. 3). The extensive new NMR data presented here adds to our understanding of this region.

Response:

We agree with the Reviewer that structural variability in the MIZ1 BTB domain has already been discussed in the literature. Our NMR data demonstrate that MIZ1 BTB domain adopts closed conformation in solution, which is a new result. Moreover, with comprehensive spin relaxation data we demonstrate that the B4/B2 region of MIZ1 BTB experiences slow μ s-ms motions, which may contribute to the presence of structural variability observed in the crystal structures.

Reviewer #2

Excellent paper - really nice piece of work, well thought out experiments to follow up an interesting observation. The correlation between backbone dynamics, FBS output and cryptic sites is a really interesting one, and will be of interest to the wider community. My only negative comments are rather trivial - there is a typo on p8 ("trough space" rather than "through space") and on p21 the authors may mean "13.3ms constant time" rather than "contact time" which usually refers to ssNMR). Apart from these minor points, the manuscript is well written, well reasoned and clear.

Response:

We thank the Reviewer for appreciating our work, and we corrected these typos.

Reviewer #3

Major points:

*1. The authors show and discuss the excellent correlation between dynamics and ligand binding. Yet, I do not think that they manage to demonstrate causation. The sentence on page 15 "**This finding suggests that protein dynamics, but not structural features, govern the ligandability of BTB domains**" does not lay on solid enough ground. First, the authors present no physical model that could explain why dynamics leads to ligand binding. Second, if I may clearly state a hypothesis, which surprisingly appears in the abstract but not anywhere else in the manuscript: the most likely explanation is that the conformational exchange identified is between a major state unable to bind and a minor state that exposes a so-called cryptic pocket, that is much more prone to ligand binding. Determining the structure of such a minor state is difficult and only a few investigations so far have been able to determine such conformations.*

However, I can suggest a few investigations that could be done to probe this hypothesis. For instance, if the minor state is similar to the bound conformation to HUWE1, the authors could compute the ligandability of the structures of MIZ1_BT B free and bound (without HUWE1 of course). Have the authors tried to co-crystallize MIZ1_BT B and 2GG4? This would definitely identify the putative cryptic pocket. The authors may also suggest other approaches. To conclude on this point, if the authors were to show that a cryptic pocket is only exposed in a minor conformational state in MIZ1_BT B, this would be a fantastic result, of interest to a very wide audience.

Response

We thank the Reviewer for suggestions to better explain the presence of dynamics and the mechanism of ligand binding to MIZ1. To improve our understanding of the role of dynamics in ligand binding to MIZ1, we have performed the following studies and made changes in the manuscript:

- we determined that the MIZ1 BT B domain predominantly adopts "closed" conformation in solution (Supplementary figure 4); the presence of the μ s-ms dynamics indeed suggests that apo MIZ1 may sample other conformations such as those found in crystal structures (see Supplementary Figure 3) or in the complex with HUWE1.

- determination of the structure of a minor state is indeed challenging, and to better understand the role of dynamics in the MIZ1 BT B domain, we have characterized the dynamics for the MIZ1-HUWE1 complex. Despite experimental challenges (loss of the BT B domain symmetry in MIZ1-HUWE1 leading to doubling of the resonances, need to perdeuterate the MIZ1 protein), we have obtained accurate relaxation parameters (Supplementary Figure 9) and found that binding of HUWE1 ligand quenched the μ s-ms dynamics but not the fast dynamics in the MIZ1 BT B domain within the B4 motif (Figure 5D, Supplementary Figure 10, 119). This very relevant finding supports the functional role of the slow μ s-ms motions in the MIZ1 BT B domain in ligand binding. Such dynamics may indeed generate potential cryptic pockets suitable for ligand binding. We have added the new data describing dynamics in MIZ1-HUWE1 as a new paragraph.

Of note, we have performed the spin relaxation dynamics studies with HUWE1, not 2GG4, due to two orders of magnitude stronger binding affinity of the HUWE1-derived peptide ligand needed to obtain a fully saturated complex with MIZ1.

- as suggested by the Reviewer, we have computed the ligandability of apo MIZ1 and MIZ1 from the complex with HUWE1 using FTMAP software. We found fewer ligand binding hot spots in the apo BT B domain (Supplementary Figure 12). On the contrary, FTMAP predicted an additional one hot spot on the interface with HUWE1 (circled in Supplementary Figure 12). This prediction is consistent with the Reviewer's comments, and we have added a brief comment to the discussion in the revised manuscript.

- we have attempted to co-crystallize MIZ1 BT B with several small molecule fragments, but, unfortunately, we failed to obtain co-crystal structures with fragments and instead, we have determined the crystal structure of apo MIZ1 BT B in the "open" conformation, which has been included in this manuscript (see Supplementary Table 1).

- we have also added a brief discussion that the "presence of μ -ms dynamics in solution may reflect the presence of cryptic pockets suitable for ligand binding, which may be explored for design of small molecule inhibitors."

2. In the case of KAISO BTB, the elevated nitrogen-15 transverse relaxation rates (and the need for deuteration) point to multimers larger than dimers. In the case of KAISO but also all other BTB domains, could the authors investigate the multimers present in solution: are overall correlation times from the model-free analysis in agreement with the dimeric form for MIZ1 and LRF? Is KAISO_BTBTB in equilibrium between a dimer and a dimer of dimer? What are the exposed surfaces in dimers or larger multimers? These questions are essential when comparing ligandability.

Response

This is an excellent point, and to better understand oligomerization status, we have performed a detailed analysis of the total correlation times for the BTB domain proteins. The correlation times for LRF and MIZ1 are, respectively, 16.7 and 16.1 ns, which corresponds to BTB domain dimers (Supplementary Figure 7). In the case of KAISO, we have noted a higher correlation time of 22.4 ns, which can be explained by the presence of a more disordered C-terminus of KAISO-BTB, decreasing the tumbling in solution. Of note, the predicted correlation time for the BTB domain tetramer is ~ 36 ns, which is significantly larger than the correlation time determined for the KAISO BTB. We have also determined the correlation time for the somewhat larger MIZ1-HUWE1 complex (30.3 kDa, $\tau_R = 24.1$ ns) and found that its correlation time is also longer than for the KAISO BTB domain (29.0 kDa, $\tau_R = 22.4$ ns). To independently validate the dimerization states of these BTB domain proteins, we performed the size exclusion analysis at the same concentrations as used in the spin relaxation NMR experiments. We found that all these proteins are clearly dimeric with less than 0.4 % of tetramers (Supplementary Figure 7). In summary, our data demonstrate that LRF, KAISO and MIZ1 BTB domains are dimeric in solution, and thus, they all have similar interfaces that might be involved in ligand binding.

Minor points:

- Could the authors discuss the solubility of the fragments presented in Figure 1?

Response: This is an important question, and we carefully tested the solubility of all compounds in our libraries. Most of the compounds are soluble at 1 mM concentration in a buffer with 5% DMSO (v/v). In addition, we also routinely calibrate the concentrations of all compounds used for the NMR titrations measuring the 1D ^1H NMR spectra with the reference compound. We have added such NMR calibration experiments in Supplementary Figure 1, showing very good solubility and accurate concentrations of fragments used for the titrations with the MIZ1 BTB domain.

- In Figures 2A-B and 2C, it is somehow difficult to compare the different views. Could the authors present the structure of the complex (2C) in the same orientation as panels 2A and 2B?

Response

- We have revised Fig 2 and mapped the chemical shift perturbations on the structures of MI1 BTB domain dimers in "open" and "closed" conformations presented in the same orientation as the structures in panel b.

- *With respect to fast motions, it is clear in Figure 4A that the beta4 strand is particularly dynamic in MIZ1. However, the picture is more contrasted for beta2, where only a handful of residues show slightly lower order parameters. This should be reflected in the way the authors discuss these data.*

Response

We agree with the Reviewer, and we have made revisions in the text to emphasize that this is particularly the beta 4 fragment that experiences the elevated dynamics (pages 12, 14 and 15).

- *Page 12, the authors write "a strong contribution from the chemical exchange (Rex)", a more rigorous phrasing would be "a contribution of chemical exchange to transverse relaxation (Rex)"*

Response

This has been corrected as suggested by the Reviewer

REVIEWER COMMENTS

Reviewer #1 (Remarks to the Author):

The revised MS makes several corrections/adjustments to the text and includes new NMR experiments. Overall, the authors present a beautiful analysis of dynamics at the Miz1 ligand-binding hotspot and the inclusion of new relaxation data make the point of flexibility at the binding site even stronger. This was never in question.

This MS presents a nice example of a cryptic binding site. Cryptic sites are, by definition, dynamic, and it would have been nice to mention the efforts from many labs to include the flexibility of binding pockets in binding studies. Essentially all modern docking programs incorporate some degree of binding site flexibility. The Vajda lab (the authors of FTMap, which is used extensively in this MS) have recently released FTMove (Egbert et al, JMB 2022), an enhancement of their original method that accounts for target flexibility by extracting additional relevant PDB structures for hotspot identification – the idea being that multiple static PDB structures represent samples of the accessible conformational space of the protein, and this can reveal additional binding sites.

The point is that protein dynamics have long been known to be important for PPIs and small molecule binding. The revised text is an improvement from the original submission in that it partially tones down the novelty of the result. The present MS is an excellent case study, and can stand on this strength without having to make overly broad generalizations, or by overlooking previous work on the role of protein dynamics in ligandability.

For example, the Title states that increased protein dynamics *defines* ligandability. As a counterexample, there are several examples of BTB domains with large, disordered loops that are unlikely to be ligand binding sites (e.g. human ZBTB4 and HIC1). Additionally, single point mutations can abrogate ligand binding without affecting dynamics. Overall, a more balanced presentation would make this MS even stronger.

Minor corrections:

Suppl. Figure 4. The relevant PDB ID is 2Q81, not 28Q1. Also, panel A appears to show 2Q81:B (not A). Panel C: left is 2q81:B (closed) right is 2q81:A (open).

Suppl. Figure 13. As mentioned in my earlier review, the SMRT peptide does not bind to Miz1 or Kaiso. Other peptides may bind to the site but none are currently known. This Figure should be deleted. Even as a "concept" diagram, it adds very little to the manuscript.

Reviewer #3 (Remarks to the Author):

The revised version of the manuscript by Kharchenko et al. reads much better. The content and conclusions are much more in line with less bold claims and more complete work. Very good work from the authors! I have just a handful of small corrections to suggest before this manuscript is ready for publication.

Abstract:

I would probably change the "can identify" with "can contribute to the identification" or "is essential to" or "can assist the identification". Overall, the idea is that NMR is important to identify cryptic pockets but unambiguously identifying a cryptic pocket will be difficult with NMR alone and likely require a combination of methods.

Discussion:

Lines 302-304: In the sentence: "We found that MIZ1^{BTB} has a uniquely elevated dynamics profile..." The authors should be careful, as they mix up in the same sentence, micro-millisecond dynamics (R_{ex}) and pico-nanosecond dynamics (S₂). The best would be to write two sentences, one about ps-ns dynamics and the other one about us-ms dynamics.

Line 310: the authors write that the quenching of u - m s dynamics in $MIZ1^{\{BTB\}}$ upon HUWE1 binding suggests "that slow dynamics plays an important role in ligand binding". This is a bit of a shortcut. I would probably reorganize this and the next few sentences in order to have the following sequence: (1) u - m s dynamics is quenched upon HUWE1 binding, (2) it is likely that HUWE1 selects a conformation from $MIZ1^{\{BTB\}}$, (3) there is evidence that the conformation of $MIZ1^{\{BTB\}}$ in the bound form is different from the dominant conformation in the free form, (4) conclusion from lines 314-315: this "may reflect the presence of cryptic pockets", (5) highlighting the important role of u - m s dynamics in ligand binding.

We would like to thank the reviewers for the additional time and work spent on our manuscript.

Detailed point-by-point answers are below:

“Reviewer #1 (Remarks to the Author):

The revised MS makes several corrections/adjustments to the text and includes new NMR experiments. Overall, the authors present a beautiful analysis of dynamics at the Miz1 ligand-binding hotspot and the inclusion of new relaxation data make the point of flexibility at the binding site even stronger. This was never in question.”

Thank you for this note of appreciation.

“This MS presents a nice example of a cryptic binding site. Cryptic sites are, by definition, dynamic, and **it would have been nice to mention the efforts from many labs to include the flexibility of binding pockets in binding studies**. Essentially all modern docking programs incorporate some degree of binding site flexibility. The Vajda lab (the authors of FTMap, which is used extensively in this MS) have recently released FTMove (Egbert et al, JMB 2022), an enhancement of their original method that accounts for target flexibility by extracting additional relevant PDB structures for hotspot identification – the idea being that multiple static PDB structures represent samples of the accessible conformational space of the protein, and this can reveal additional binding sites.”

We thank the Reviewer for this comment, and we have explored the new FTMove software developed by Vajda lab to evaluate BTB domains. However, we found that by including additional variability in the crystal structures, FTMove predicts even more small molecule binding hot spots in proteins we found undruggable based on fragment screening results. For example, predictions for the LRF BTB domain shows multiple hot spots determined by FTMove:

Figure 1. Comparison of hot spot prediction in LRF BTB domain using FTMap (left) and FTMove (right).

Therefore, we decided not to add FTMove analysis in the revised manuscript.

“The point is that protein dynamics have long been known to be important for PPIs and small molecule binding. The revised text is an improvement from the original submission in that it partially tones down the novelty of the result. **The present MS is an excellent case study, and can stand on this strength** without having to make overly broad generalizations, or by overlooking previous work on the role of protein dynamics in ligandability.”

Thank you for the positive note and for emphasizing the development of computational methods incorporating dynamics to predict ligandability; we have added an additional comment in the manuscript (see page 3 in the introduction) and associated references.

“For example, the Title states that increased protein dynamics *defines* ligandability. As a counterexample, there are several examples of BTB domains with large, disordered loops that are unlikely to be ligand binding sites (e.g. human ZBTB4 and HIC1). Additionally, single point mutations can abrogate ligand binding without affecting dynamics. Overall, a more balanced presentation would make this MS even stronger.”

We agree with this comment and to more precisely describe the findings in our manuscript, we have changed the title to “Increased slow dynamics defines ligandability of BTB domains”.

To make a more balanced presentation, we also made other changes to the manuscript:

1. We changed the last sentence in the abstract to:

“Our data argue that examining protein dynamics using NMR can contribute to the identification of cryptic binding sites and may support the prediction of the ligandability of novel challenging targets”.

2. We have changed the last sentence in the discussion to more precisely emphasize the role of slow dynamics in the prediction of druggability:

“We anticipate that elevated μ s-ms dynamics may represent a general feature of druggable proteins ...”

Minor corrections:

Suppl. Figure 4. The relevant PDB ID is 2Q81, not 28Q1. Also, panel A appears to show 2Q81:B (not A). Panel C: left is 2q81:B (closed) right is 2q81:A (open).

Thank you for spotting this. We corrected the typos.

Suppl. Figure 13. As mentioned in my earlier review, the SMRT peptide does not bind to Miz1 or Kaiso. Other peptides may bind to the site but none are currently known. This Figure should be deleted. Even as a “concept” diagram, it adds very little to the manuscript.

We removed the figure.

Reviewer #3 (Remarks to the Author):

The revised version of the manuscript by Kharchenko et al. reads much better. The content and conclusions are much more in line with less bold claims and more complete work. Very good work from the authors! I have just a handful of small corrections to suggest before this manuscript is ready for publication.

Thank you so much!

Abstract:

I would probably change the “can identify” with “can contribute to the identification” or “is essential to” or “can assist the identification”. Overall, the idea is that NMR is important to identify cryptic pockets but unambiguously identifying a cryptic pocket will be difficult with NMR alone and likely require a combination of methods.

We have made the requested change to:

“Our data argue that examining protein dynamics using NMR can contribute to the identification of cryptic binding sites, and may support the prediction of the ligandability of novel challenging targets.”

Discussion:

Lines 302-304: In the sentence: “We found that MIZ1^{BTB} has a uniquely elevated dynamics profile...” The authors should be careful, as they mix up in the same sentence, micro-millisecond dynamics (R_{ex}) and pico-nanosecond dynamics (S_2). The best would be to write two sentences, one about ps-ns dynamics and the other one about us-ms dynamics.

We have revised the text as suggested by the Reviewer.

Line 310: the authors write that the quenching of us-ms dynamics in MIZ1^{BTB} upon HUWE1 binding suggests “that slow dynamics plays an important role in ligand binding”. This is a bit of a shortcut. I would probably reorganize this and the next few sentences in order to have the following sequence: (1) us-ms dynamics is quenched upon HUWE1 binding, (2) it is likely that HUWE1 selects a conformation from MIZ1^{BTB}, (3) there is evidence that the conformation of MIZ1^{BTB} in the bound form is different from the dominant conformation in the free form, (4) conclusion from lines 314-315: this “may reflect the presence of cryptic pockets”, (5) highlighting the important role of us-ms dynamics in ligand binding.

This text has been revised to incorporate the Reviewers’ suggestions.